# DiffuGesture: Generating Human Gesture From Two-person Dialogue With Diffusion Models

Weiyu Zhao
Harbin Institute of Technology
Weihai, Shandong, China
weiyuzhao66@gmail.com

Liangxiao Hu*
Harbin Institute of Technology
Weihai, Shandong, China
lx.hu@hit.edu.cn

Shengping Zhang
Harbin Institute of Technology
Weihai, Shandong, China
s.zhang@hit.edu.cn

## ABSTRACT

This paper describes the DiffuGesture entry to the GENEA Challenge 2023. In this paper, we utilize conditional diffusion models to formulate the gesture generation problem. The DiffuGesture system generates human-like gestures from the two-person dialogue scenario, which are responsive to the interlocutor motions and accompany with the input speech. DiffuGesture system is built upon the recent DiffGesture [39]. Specifically, we introduce a lightweight transformer encoder to fuse the temporal relationships between human gestures and multi-modal conditions. Moreover, we adopt implicit classifier-free guidance to trade off between diversity and gesture quality. According to the collective evaluation released by GENEA Challenge 2023, our system demonstrates strong competitiveness in the appropriateness evaluation.

## CCS CONCEPTS

• **Computing methodologies** → **Animation**; **Neural networks**;
• **Human-centered computing** → *Virtual reality*.

## KEYWORDS

gesture generation, diffusion models, neural networks

**ACM Reference Format:**
Weiyu Zhao, Liangxiao Hu*, and Shengping Zhang. 2023. DiffuGesture: Generating Human Gesture From Two-person Dialogue With Diffusion Models . In *INTERNATIONAL CONFERENCE ON MULTIMODAL INTERACTION (ICMI '23 Companion), October 9–13, 2023, Paris, France.* ACM, New York, NY, USA, 7 pages. https://doi.org/10.1145/3610661.3616552

## 1 INTRODUCTION

Human gestures serve as a distinct mode of communication in daily conversations, which assists the speakers in conveying semantic information more effectively and facilitates interpersonal communication. [21, 29]. Therefore, generating realistic co-speech human gestures from conversations plays a crucial role in achieving improved interaction between virtual entities and humans. Our goal

*Corresponding author.

is to generate co-speech human gestures from the two-person dialogue. However, generating human gestures with multi-modal data such as audio, text, and conversational cues in two-person dialogue remains a challenging and unresolved problem.

Early research in data-driven co-speech gesture generation approaches often relies on statistical analysis. Levine [16] et al. utilize probabilistic models to establish the relationship between audio and gestures. In recent years, deep learning methods have been increasingly applied in co-speech gesture generation. Kucherenko [12] et al. and Yoon [34] et al. employ the multi-layer perceptron (MLP) and recurrent neural network (RNN) methods to generate deterministic human gestures, respectively. However, these approaches do not adequately address the implicit mapping between the data and gestures [13]. To achieve more diverse and personalized gesture movements and improve the mapping between data and gestures, there emerge methods using GAN [3, 25, 30], diffusion models [27, 32, 39] and VQ-VAE [20, 22].

However, these methods mainly focus on single-person co-speech gesture generation. In this paper, we present a novel approach for co-speech human gesture generation in the two-person dialogue scenario. Specifically, given the behavior of the interlocutor and the audio and textual transcriptions of the main agent, we generate the reaction and co-speech movements of the main agent, respectively. Inspired by [39], we adopt conditional diffusion models for co-speech gesture generation from the two-person dialogue. Specifically, we introduce a lightweight transformer encoder to enhance the contextual relevance between human gestures and multi-modal conditions. Finally, we introduce implicit classifier-free guidance to trade off between diversity and gesture quality.

The main contributions of our work are:

- We present an early attempt to utilize conditional diffusion models for co-speech human gesture generation from two-person dialogue, which generates impressive co-speech gesture movements.
- We introduce a lightweight transformer encoder that effectively fuses the temporal relationships between human gestures and multi-modal conditions.

## 2 RELATED WORK

In this section, we will discuss the previous work in the fields of gesture generation and diffusion model generation.

### 2.1 Data-driven Gesture Generation

The data-driven approach to gesture generation has found extensive applications across various domains.In recent years, researchers have utilized audio [6, 17, 18, 22], transcribed text [3, 10, 23, 26, 27, 36], and multimodal data [2, 19, 33] to drive gesture generation. The

use of audio-driven gesture generation is quite common in various applications. For example, Ginosaret et al. [6] utilize an adversarial discriminator to regress gestures from audio. Qian et al. [22] employ conditional learning to achieve audio-driven gesture generation, alleviating the ambiguity in simultaneous speech and gesture synthesis. Audio2gestures [18] and DanceFormer [17] use a variational autoencoder [11] and Transformer [28], respectively, to generate gestures from audio. Text-driven motion synthesis can be seen as learning a joint embedding of the text feature space and the motion feature space[22]. Text2gestures [3] establishes the connection between text and gesture actions using a transformer. T2M-GPT [36] and MotionGPT[10], built upon generative pre-trained transformer (GPT), treat gesture actions as a language and utilize VQ-VAE to transform text into gesture actions. MDM [27] and MotionClip [26] preprocess transcribed text using CLIP[23] to establish the conversion between action and text embeddings.

Recently, there has been an increasing trend in co-speech gesture generation to use multimodal data, including audio, text, and speaker ID. Yoon et al. [33] proposed a model that combines multimodal context and adversarial training to generate gestures that resemble human-like movements and are synchronized with the speech content and rhythm. Rhythmic Gesticulator [2] is the first model to use neural networks to establish the relationship between gestures and audio in terms of rhythm and semantics. HA2G [19] leverages contrastive learning strategies to fully utilize the rich connections between speech audio, text, and human gestures, resulting in the generation of realistic gesture movements. However, none of the aforementioned works considered the influence of other individuals in dyadic conversations on the embodied agents.

## 2.2 Diffusion Models

Diffusion models are a type of probabilistic generative model based on stochastic processes [8], where initial data points gradually evolve towards the target distribution through a diffusion process at each time step. Dhariwal et al. [5] introduce classifier guidance to improve sample quality and generate higher-quality results. Then, the introduction of the Classifier-Free Guidance [9] eliminates the need for explicit classification models and supports more openended and exploratory generation in various tasks. Diffusion models have recently been widely applied in various fields, such as image generation [24], 3D shape generation [31], video generation [7].

More recently, in the context of gesture generation tasks, diffusion generative models [1, 27, 37, 39] have also been employed for co-speech gesture generation. Inspired by the work of DiffGesture [39] in 2D gesture generation, we have developed a framework for generating 3D gesture poses from multimodal data in a twoperson dialogue scenario.

## 3 METHOD

Given the behavior of the interlocutor and the audio and textual transcriptions of the main agent, our goal is to generate the listening reactions and co-speech motions simultaneously. The architecture of our system is depicted in Figure 1(a). We first introduce the problem definition in Section 3.1. Then we present the diffusion process and reverse process for gesture generation in Section 3.1. Finally, we develop a transformer encoder to fuse the temporal

relationships between human gestures and multi-modal conditions in Section 3.3.

## 3.1 Problem Definition

Given the sequences of 3D full-body motions, we represent them as $x = \{p_1, p_2, p_3, ..., p_n\} \in \mathbb{R}^{N \times 3J}$, $N$ represents the sequence length and $J$ denotes the total joint number. The reverse denoising process $G$ of the diffusion model is parameterized by $\theta$ to synthesize the main agent skeleton sequence $x_m$, which is further conditioned on the multi-modal conditions $C$ and the initial poses of the previous $M$ frames $x_{pre}$. The learning objective can be expressed as $argmin_\theta \left\| x_m - G_\theta(C, x_{pre}) \right\|$.

## 3.2 Diffusion-based Gesture Generation

Inspired by the previous work [39], we extend this model in the two-person dialogue scenario. Unlike generating 2D skeletal upperbody poses in [39], we synthesize the full-body human gestures in a two-person dialogue scenario.

**Diffusion Process.** The diffusion process, also known as the forward process, is used to approximate the posterior distribution $q(x_{1:T}|x_0)$. It gradually introduces Gaussian noise into the original distribution based on the variance sequence $\beta_1, ..., \beta_t$, where $\beta_i \in (0, 1)$. The diffusion process is defined as follows:

$$q(x_t^{1:N}|x_{t-1}^{1:N}) = \mathcal{N}(\sqrt{\beta_t}x_{t-1}^{1:N}, (1 - \beta_t)I), \tag{1}$$

$$q(x_{1:T}|x_0) = \prod_{t=1}^{T} q(x_t^{1:N}|x_{t-1}^{1:N}), \tag{2}$$

where $x_t^{1:N}$ represents the main agent motion sequence $\{p_m\}_{i=1}^{N}$ at $t$ denoising step. Next, we will slightly abuse the use of letters and use $x$ to represent $x^{1:N}$. By progressively adding noise in this manner to the original gesture motions $x_0$, it approaches a distribution that closely resembles white noise.

**Reverse Process.** The reverse process, also known as the generation process, estimates the joint distribution $p_\theta(x_{0:T})$. The reverse process of diffusion models also maintains the form of Gaussian transition. Additionally, following the idea of classifier-free guidance, we train the model in both unconditional and conditional generation settings to generate more realistic and diverse gesture motions. The reverse process is defined as follows:

$$p_\theta(x_{0:T}) = p_\theta(x_T) \prod_{t=1}^{T} p_\theta(x_{t-1}|x_t, C), \tag{3}$$

$$where \quad p_\theta(x_{t-1}|x_t, C) = \mathcal{N}(x_{t-1}; \mu_\theta(x_t, t, C), \sum_\theta(x_t, t)). \tag{4}$$

Equation 4 represents the conditional generation and we set the conditions $C$ as zero (denoted as $\phi$) for unconditional generation in the training stage. The corrupted noisy gesture sequence $x_t$ is sampled by $q(x_t|x_0)$.

**Traning loss.** According to DDPM [8], the previous corrupted gesture sequence $x_{t-1}$ is defined as follows:

$$x_{t-1} = \frac{x_t - \sqrt{1 - \bar{\alpha_t}}\hat{\epsilon}}{\sqrt{\bar{\alpha_t}}}, \tag{5}$$

$$where \quad \bar{\alpha_t} = \prod_{i=1}^{t} 1 - \beta_i. \tag{6}$$

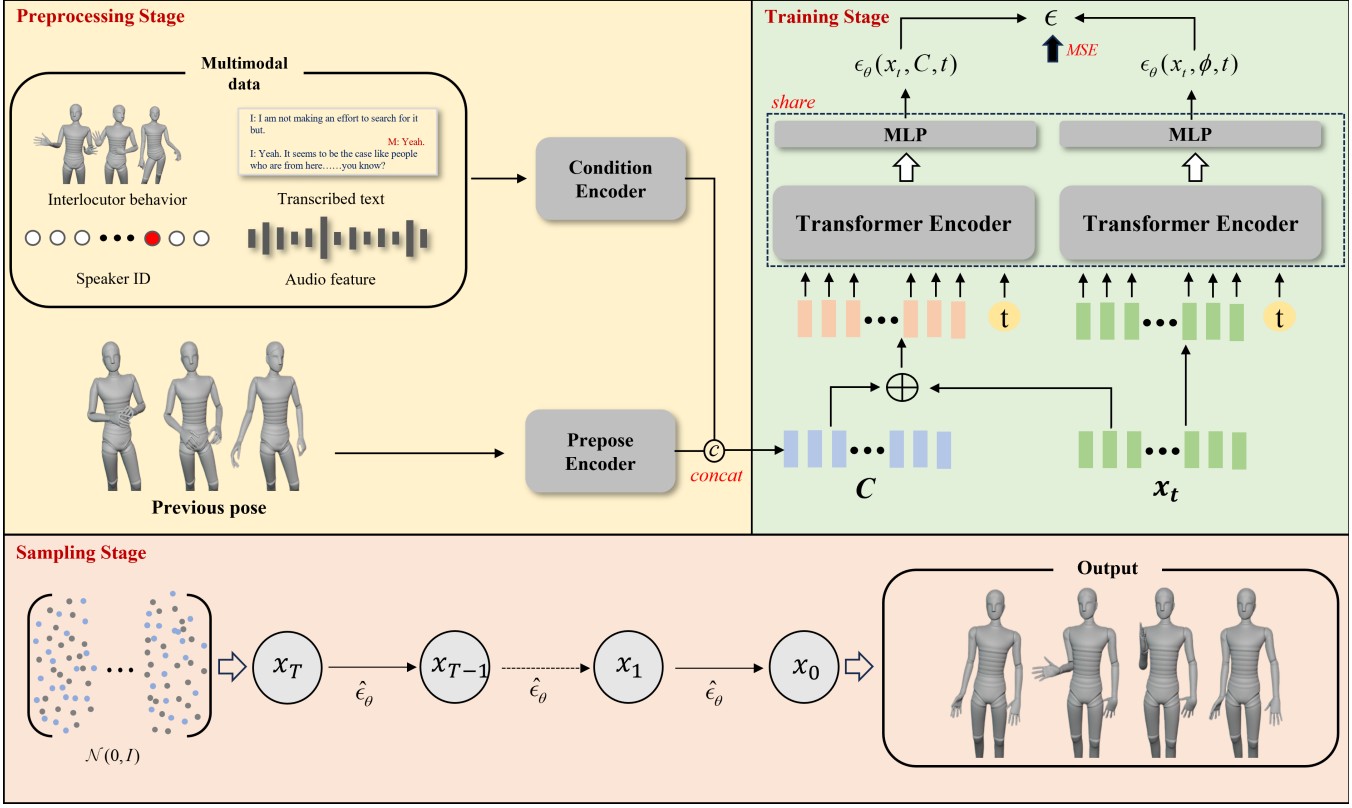

**Figure 1: Overview of the Diffu2guesture framwork. In the preprocessing stage (yellow), we develop a condition encoder and a propose encoder to process multi-modal data and previous poses, respectively. Then we concatenate the two outputs together to create condition features $C$. In the training stage (green), we introduce classifier-free guidance to train the transformer encoder. In the sampling stage (pink), we start with random noise $x_T$ and generate a clean sample $x_0$ through $T$ denoising steps.**

So we can denoise the Gaussian noise to the original gesture motion distribution step by step. Then, we use the Mean Squared Error (MSE) loss to compute the loss between the estimated noise and the actual noise at each time step [39]:

$$\mathcal{L}_{simple} = \mathbb{E}_q \left[ \left\| \epsilon - \epsilon_\theta(\sqrt{\bar{\alpha}_t}x_0 + \sqrt{1 - \bar{\alpha}_t}\epsilon, C, t) \right\|^2 \right]. \quad (7)$$

Where $\epsilon_\theta$ is the predicted Gaussian noise, and $\epsilon$ represents the actual added noise. During the training process, we randomly mask the conditions $C$ for the unconditional setting.

**Sampling.** Generating motion from speech is an implicit mapping rather than a direct one-to-one correspondence between speech and gestures. To ensure a better correlation between audio and actions, we introduce classifier-free guidance [5]. From the perspective of gesture generation, we can consider it as follows:

$$G_M = G(x_t, \phi, t) + s \cdot (G(x_t, C, t) - G(x_t, \phi, t)). \quad (8)$$

Where $s$ is a hyperparameter. As mentioned in the training loss section, during the training process, we utilize random masking to create unconditional input for training unconditional models. Then, we train a single transformer encoder and MLP layer under various

conditioning setups between conditional models and unconditional models. This enables us to realize classifier-free guidance.

Based on the aforementioned context, diffusion models can be used to generate natural embodied agent gestures in a two-person dialogue setting.

### 3.3 Cross-Modal Attention Encoding

Generating 3D gesture poses using conditional diffusion models is different from generating images. Both the pose sequence $x$ and the multi-modal conditions $C$ exhibit strong temporal dependencies. Here, we need to establish a module to ensure that our results are time-dependent. Unlike previous work in the GENEA 2022 challenge that utilizes LSTM [4], VQVAE [20], and graph models [38], we employ a lightweight transformer encoder to encode $N$ frames of continuous motions and multi-modal data. We align the noisy gesture sequence $x_t$ and multi-modal conditions $C$ in the time dimension and treat each frame as a separate token. The time step $t$ is treated as a separate token. We then utilize attention mechanisms for encoding.

$$Attention(Q, K, V) = softmax(\frac{QK^T}{\sqrt{d_k}})V. \quad (9)$$

Where $Q$, $K$, and $V$ are the query, key, and value matrix from input tokens, in the multi-head attention mechanism.

## 4 EXPERIMENT

### 4.1 Data Processing

The only dataset we used is the GENEA Challenge 2023 [14] dataset, which is an extension of Lee et al.'s Talking With Hands [15] dataset. The dataset includes participants consisting of a main agent (tasked with generating motion) and an interlocutor (the other party in the conversation). The conversation data in the dataset is in dyadic form, providing audio and text transcriptions for both parties, speaker IDs, and motion. In the provided official data, each recorded conversation is duplicated with flipped roles to augment the training data.

We fully leverage the various information available in the dataset, including the audio and transcribed text between the main agent and the interlocutor, as well as the speaker IDs. We follow the same processing approach as the baseline [4] for handling audio, transcriptions, and human body joints. We obtain three audio features at a sampling rate of 44100: mel-spectrograms, MFCCs, and prosodies. The frames generated have a rate of 30 FPS and their length matches the duration of the motion sequence. We encode the text using Fasttext, resulting in word vectors of dimension 300. Additionally, two extra dimensions are used to indicate whether the speaker is silent or laughing. Furthermore, we define the identity information of each speaker using one-hot encoding.

For the processing of motion data, we also select 25 joints, including the root node, which have a significant influence on skeleton motion. These joints are represented in a dimension of 78. To generate high-quality motion sequences, we segment the motion sequence into chunks of 300 frames each, which serve as inputs to the diffusion process. To ensure continuity between adjacent motion segments, we extract the preceding 50 previous poses as part of the generation condition. After aligning the audio features, encoded text, identity information, and speakers' motion sequences in the temporal dimension, we obtain the same length as the motion sequences. Similarly, the previous pose is mapped to the corresponding dimension after being processed by the prepose encoder.

### 4.2 Evaluation

The evaluation of our approach is conducted through subjective assessment by the organizers of the GENEA Challenge 2023 and other participating teams. The organizers recruit study participants residing in the UK, IE, USA, CAN, AUS, and NZ, who had English as their first language, via crowdsourcing platforms to perform the evaluations. Multiple attention checks are implemented during the experiment to ensure the participants' engagement and attentiveness. The evaluation of this challenge consisted of three aspects: **human-likeness; appropriateness for agent speech; appropriateness for the interlocutor.** The specific results are presented in Table 1 and Table 2. The natural motion is labeled NA. Our method is labeled **SB** in the tables.

**Human-likeness.** The study participants watch 8 to 10 seconds of video and rate the motion of the virtual character as human-like, independent of the dialogue content and the speaker. DiffuGesture performs poorly on this metric.

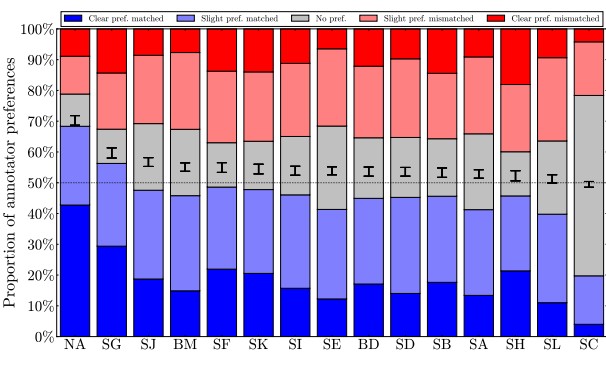

(a) Appropriateness for agent speech

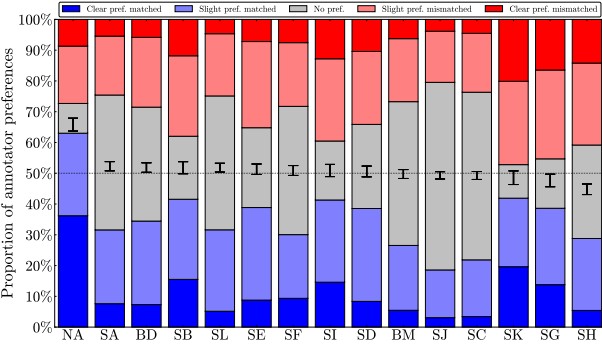

(b) Appropriateness for the interlocutor

**Figure 2: The bar plots display response distribution in appropriateness studies. The blue bar represents preferred matched motion responses, and the red bar represents preferred mismatched motion responses. The height of each bar corresponds to the fraction of responses in each category. On top of each bar is also a confidence interval for the mean appropriateness score, scaled to fit the current axes. The dotted black line indicates chance-level performance. Conditions are ordered by mean appropriateness score.**

**Appropriateness for agent speech.** This metric evaluates whether the motion of the virtual character is appropriate for the given speech while controlling for the overall human-likeness of the motion [35]. During the testing process, study participants are presented with a pair of videos, both from the same condition, where one video matches the specific speech and the other is from an unrelated speech. Both videos play the specific speech, and participants are asked to select the video they believe best matches the speech.

**Appropriateness for the interlocutor.** During the conversation process, both participants in the dialogue influence each other. Therefore, this metric evaluates whether the motion of the virtual character is appropriate for the given interlocutor's behavior (including speech and motion) while controlling for the overall human-likeness of the motion. Study participants are also presented with a pair of videos, where the behavior of the main agent remains fixed, but the behavior of the interlocutor is randomly replaced in

**Table 1: Summary statistics of user-study responses from both appropriateness studies, with confidence intervals for the mean appropriateness score (MAS) at the level $\alpha = 0.05$. "Pref. matched" identifies how often test-takers preferred matched motion in terms of appropriateness after splitting ties. Conditions are ordered by mean appropriateness score.**

### (a) Appropriateness for agent speech

| Condi- | MAS | Pref. | Raw response count | | | | | |
| tion | | matched | 2 | 1 | 0 | −1 | −2 | Sum |
|---|---|---|---|---|---|---|---|---|
| NA | 0.81±0.06 | 73.6% | 755 | 452 | 185 | 217 | 157 | 1766 |
| SG | 0.39±0.07 | 61.8% | 531 | 486 | 201 | 330 | 259 | 1807 |
| SJ | 0.27±0.06 | 58.4% | 338 | 521 | 391 | 401 | 155 | 1806 |
| BM | 0.20±0.05 | 56.6% | 269 | 559 | 390 | 451 | 139 | 1808 |
| SF | 0.20±0.06 | 55.8% | 397 | 483 | 261 | 421 | 249 | 1811 |
| SK | 0.18±0.06 | 55.6% | 370 | 491 | 283 | 406 | 252 | 1802 |
| SI | 0.16±0.06 | 55.5% | 283 | 547 | 342 | 428 | 202 | 1802 |
| SE | 0.16±0.05 | 54.9% | 221 | 525 | 489 | 453 | 117 | 1805 |
| BD | 0.14±0.06 | 54.8% | 310 | 505 | 357 | 422 | 220 | 1814 |
| SD | 0.14±0.06 | 55.0% | 252 | 561 | 350 | 459 | 175 | 1797 |
| **SB** | 0.13±0.06 | 55.0% | 320 | 508 | 339 | 386 | 262 | 1815 |
| SA | 0.11±0.06 | 53.6% | 238 | 495 | 438 | 444 | 162 | 1777 |
| SH | 0.09±0.07 | 52.9% | 384 | 438 | 258 | 393 | 325 | 1798 |
| SL | 0.05±0.05 | 51.7% | 200 | 522 | 432 | 491 | 170 | 1815 |
| SC | −0.02±0.04 | 49.1% | 72 | 284 | 1057 | 314 | 76 | 1803 |

### (b) Appropriateness for the interlocutor

| Condi- | MAS | Pref. | Raw response count | | | | | |
| tion | | matched | 2 | 1 | 0 | −1 | −2 | Sum |
|---|---|---|---|---|---|---|---|---|
| NA | 0.63±0.08 | 67.9% | 367 | 272 | 98 | 189 | 88 | 1014 |
| SA | 0.09±0.06 | 53.5% | 77 | 243 | 444 | 194 | 55 | 1013 |
| BD | 0.07±0.06 | 53.0% | 74 | 274 | 374 | 229 | 59 | 1010 |
| **SB** | 0.07±0.08 | 51.8% | 156 | 262 | 206 | 263 | 119 | 1006 |
| SL | 0.07±0.06 | 53.4% | 52 | 267 | 439 | 204 | 47 | 1009 |
| SE | 0.05±0.07 | 51.8% | 89 | 305 | 263 | 284 | 73 | 1014 |
| SF | 0.04±0.06 | 50.9% | 94 | 208 | 419 | 208 | 76 | 1005 |
| SI | 0.04±0.08 | 50.9% | 147 | 269 | 193 | 269 | 129 | 1007 |
| SD | 0.02±0.07 | 52.2% | 85 | 307 | 278 | 241 | 106 | 1017 |
| BM | −0.01±0.06 | 49.9% | 55 | 212 | 470 | 206 | 63 | 1006 |
| SJ | −0.03±0.05 | 49.1% | 31 | 157 | 617 | 168 | 39 | 1012 |
| SC | −0.03±0.05 | 49.1% | 34 | 183 | 541 | 190 | 45 | 993 |
| SK | −0.06±0.09 | 47.4% | 200 | 227 | 111 | 276 | 205 | 1019 |
| SG | −0.09±0.08 | 46.7% | 140 | 252 | 163 | 293 | 167 | 1015 |
| SH | −0.21±0.07 | 44.0% | 55 | 237 | 308 | 270 | 144 | 1014 |

**Table 2: Summary statistics of user-study ratings from the human-likeness study, with confidence intervals at the level $\alpha = 0.05$. Conditions are ordered by decreasing sample median rating. Our entry is SB.**

| Condi- | Human-likeness | |
| tion | Median | Mean |
|---|---|---|
| NA | 71 ∈ [70, 71] | 68.4±1.0 |
| SG | 69 ∈ [67, 70] | 65.6±1.4 |
| SF | 65 ∈ [64, 67] | 63.6±1.3 |
| SJ | 51 ∈ [50, 53] | 51.8±1.3 |
| SL | 51 ∈ [50, 51] | 50.6±1.3 |
| SE | 50 ∈ [49, 51] | 50.9±1.3 |
| SH | 46 ∈ [44, 49] | 45.1±1.5 |
| BD | 46 ∈ [43, 47] | 45.3±1.4 |
| SD | 45 ∈ [43, 47] | 44.7±1.3 |
| BM | 43 ∈ [42, 45] | 42.9±1.3 |
| SI | 40 ∈ [39, 43] | 41.4±1.4 |
| SK | 37 ∈ [35, 40] | 40.2±1.5 |
| SA | 30 ∈ [29, 31] | 32.0±1.3 |
| **SB** | 24 ∈ [23, 27] | 27.4±1.3 |
| SC | 9 ∈ [ 9, 9] | 11.6±0.9 |

one of the videos. Participants are then asked to select the video that best matches the behavior of the interlocutor. DiffuGesture has achieved promising results in this metric.

## 5 DISCUSSION

As shown in Table 1, we achieve satisfactory results in both metrics of appropriateness for agent speech and the interlocutor. Our scores for these two metrics are 0.13 and 0.07, respectively. For the

appropriateness of the interlocutor, we achieve favorable results. The score of the "Preferred Match" category is 51.8%. Furthermore, as shown in Figure 2(b), a considerable proportion of participants chose our results as their preferred matched motion responses. We believe that several factors contribute to these results. Firstly, we make effective use of the provided information, including audio, transcribed text, and interlocutor behavior. Our data processing methods have demonstrated their effectiveness. Additionally, the introduced cross-modal attention encoder proves to be effective. It enables us to adequately encode information from different modalities, thus generating plausible motions of the main agent with respect to the behavior of the interlocutor.

We also achieve unsatisfactory results in the human-likeness metric, with a score of only 24. The challenge provides long-term human gesture sequences with variable lengths. Our naive diffusion models without specific designs only support generating fixed-length motion sequences. We segment the condition sequences and simply predict 300 frames for each segment and concatenate the predicted fixed-length motion sequences to generate the complete motions. This results in noticeable jitter at the junctions of the predicted fixed-length motion sequences. To eliminate the phenomenon, we also make some effort such as taking the previously predicted motions and acceleration between adjacent frames as part of the conditions. Furthermore, we also increase the length of generated sequences to reduce the discontinuities of generated motions. However, these naive methods do not yield the expected results. The acceleration constraint reduces the richness of the generated motions, making them less human-like. We also mention that the provided motion sequences for evaluation are not final optimized ones. This may cause undesired evaluation results.

# 6 CONCLUSION

We propose the DiffuGesture as described in this paper to participate in the GENEA Challenge 2023. Based on conditional diffusion models, we develop a system that generates co-speech human gestures for the main agent in the two-person dialogue. In our system, we encode the features of audio, transcriptions, interlocutor behavior using a transformer encoder. Furthermore, we adopt classifier-free guidance to trade off between diversity and gesture quality. The evaluation results show that DiffuGesture performs well in terms of appropriateness for the interlocutor metric. However, compared to other systems participating in the challenge, it does not generate high-fidelity human-like motions effectively.

In the future, we will continue to explore conditional diffusion models to generate high-fidelity co-speech human gestures in various scenarios. We aim to handle the generation of variable-length motion sequences and reduce the distortion of motions at breakpoints. Additionally, we intend to investigate the incorporation of semantic supervision to aid in the generation of co-speech gestures. We will focus on these aspects in our future work.

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

## A RESEARCH METHODS

### A.1 Significant differences for Appropriateness

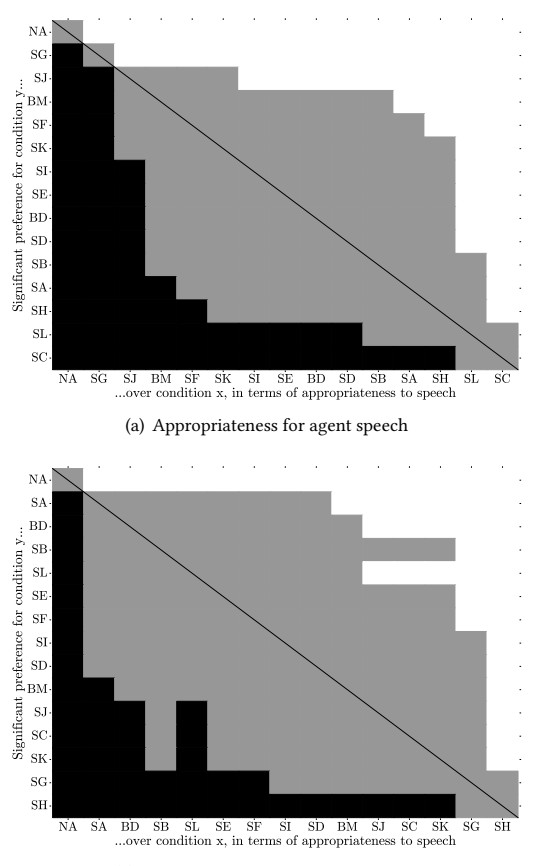

(a) Appropriateness for agent speech

(b) Appropriateness for the interlocutor

**Figure 3: Significant differences between conditions in the two appropriateness studies. White means the condition listed on the y-axis achieved an MAS significantly above the condition on the x-axis, black means the opposite (y scored below x), and grey means no statistically significant difference at level $\alpha = 0.5$ after correction for the false discovery rate.**

### A.2 Significant differences for Human-likeness

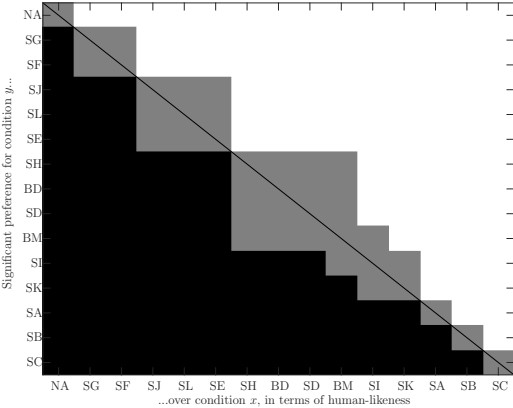

(a) Appropriateness for agent speech

**Figure 4: Significance of pairwise differences between conditions. White means that the condition listed on the $y$-axis rated significantly above the condition on the $x$-axis, black means the opposite ($y$ rated below $x$), and grey means no statistically significant difference at the level $\alpha = 0.05$ after Holm-Bonferroni correction.**