# OpenReview forum: "DiffuGesture: Generating Human Gesture From Two-person Dialogue With Diffusion Models"
_ACM.org/ICMI/2023/Workshop/GENEA_Challenge — GENEA Challenge 2023 Workshopproceeding_

### Official Review · Reviewer_1VjV · 2023-07-19
**Review of Diffu2Gesture**

**Rating:** 6
**Confidence:** 4

**Review:**

Paper Summary:

The paper presents a novel approach to gesture generation using diffusion models. The authors extend the work from the upper body to the full body, which seems to have resulted in a decrease in performance compared to the original DiffGesture. The paper discusses the use of shared MLP and Transformer models, but the specifics are not clearly explained.

Relevance:

The paper is relevant to the field of gesture generation, particularly in the context of using diffusion models. The extension of the work from upper body to full body is a significant contribution, despite the observed decrease in performance.

Significance:

The paper's significance lies in its novel approach to gesture generation using diffusion models. However, the results indicate that the quality of the generated gestures is not as good as expected, which raises questions about the effectiveness of the proposed method.

Paper Strengths:

The paper presents a novel approach to gesture generation using diffusion models.
The extension of the work from the upper body to the full body is a significant contribution.

Paper Weaknesses:

The performance of the proposed method seems to be lower than the original DiffGesture.
The paper could benefit from a more detailed explanation of the proposed method, particularly the shared MLP and Transformer models.
The explanation of the principle, specifically the balance between diversity and quality, is not clear.
The results indicate that the quality of the generated gestures is not as good as expected. The authors should discuss why the quality of the generated gestures is not as good as expected, despite the use of diffusion models.

Further Comments:

1. the result seems not as good as the original DiffGesture, is it a problem because the original upper body work is extended to the whole body? 2. the specific method is not clearly stated, it is too simple, what is the meaning of share in Fig. 1, is MLP and Transformer the same one? 3. the principle is not clearly explained, the L276 is supposed to be in the conditional control for equilibrium, not diversity and quality? Diversity is controlled by the different noises of the inputs, and quality is determined by the architecture of the model itself. 4. 4. L535 According to Figure 3(a) of Appendix A1, besides NA, there are also SG and SJ which are significantly more than the proposed system? 5. L438 What do you mean by per second, is it per chunk or per segment? 6. L443 [300,932] Consider a different way of writing it, it looks like a reference.

---

### Official Review · Reviewer_RunK · 2023-07-31
**Introduces a cross-modal attention encoder to synchronize the align the pose sequence and the multi-modal conditions.**

**Rating:** 6
**Confidence:** 3

**Review:**

The paper was easy to read and well written.
The work is heavily based on a pre-existing DiffGesture by adding the ability to synchronize the temporal relashionships between human gestures and multi-modal conditions.
The literature review seems adequate.
The results seem to indicate that the pre-processing step was responsible for a reasonable good result in terms of gesture appropriateness for both the main agent and the interlocutor, however the naturalness scored quite low due to the frame generation capability of the system being capped at 300 frames.
In general it seems like the authors provide some insights as to what their method may have positively added.
However there isn't much description about how this cross-modal attention encoder actually works or how it can be replicated, only a very brief explanation in section 3.3. This seems to be a core piece of the contributions and feels like it could have received more "attention".

There are a few things I would like to see clarified in the paper if it gets accepted, before the final submission:
- Please elaborate on the limitation of 300 frames. How easy is it to remove that limit? The authors describe that they tried, however it results in a significant increase of jitter. Is there a fundamental problem with this architecture that you think may be related with that? If so, what do you think it is?

- In paragraph 5 - Discussion, the authors mention "In the appropriateness for the interlocutor metric, only the NA condition has significantly higher scores than ours." - however that seems false given that I'm looking at Table 1(b) and can see SA with a score of 53.5%, above SB's 51.8%.
- In the next sentence, they add "blue bar region indicates a higher proportion of participants choosing the correctly matched motion as their preference" - however, looking at SA in Figure 2(b), the blue bar represents about 30% of the total preference, so I don't think that the authors managed to convey what they meant.

---

### Decision · Program_Chairs · 2023-08-04

**Decision:**

Accept (Workshop proceeding)

**Comment:**

This paper received scores on the threshold of acceptance. Reviewers pointed to different bits of the manuscript (e.g., cross-modal attention encoder, the shared MLP and Transformer models, the balance between diversity and quality) as being unclear, and also wanted more information about the origin of the jitter issues seen with the system. Taking into account the totality of reviews, the organisers have decided to accept this paper to the Workshop track to be published in the Adjunct ICMI Proceedings, which gives the authors more time to improve their paper.

Please read the reviews below carefully and revise the paper for the camera-ready version. The following changes are particularly important:
* Expand and clarify Sec. 3.3
* Improve clarity regarding the shared MLP and Transformer models and the balance between diversity and quality
* Improve the discussion of the limitation to 300 frames
  - The paper states that it “is mainly because of the limitations inherent from the diffusion models”. However, diffusion processes for motion exist that generalise well to very long sequences, e.g., ref [1] in the paper. The issue cannot be with diffusion models as a model class, which the text might suggest.
  - How are sequence positions encoded (positional embeddings)? This is not mentioned in the paper. Could the issue be that the positional dependence mechanism used is not __inductive__, in the language of the overview of positional dependence mechanisms in [a]?
* Nuance the claims regarding the efficacy of the interlocutor awareness in the first paragraph of section, since (although only NA was significantly better) the mean interlocutor appropriateness score achieved was not statistically different from chance.

Side note: In paper titles and method names of the form “x2y”, “x” is usually the input to the system. In this case, however, “Diffu” stands for a method, not an input. The authors may consider whether or not this makes them wish to change the name of their method.

[a] U. Wennberg & G. E. Henter. 2021. “The Case for Translation-Invariant Self-Attention in Transformer-Based Language Models”. In Proc. ACL-IJCNLP (Volume 2: Short Papers), pp. 130–140.